# Interleukin-17A and Keratinocytes in Psoriasis

**DOI:** 10.3390/ijms21041275

**Published:** 2020-02-13

**Authors:** Masutaka Furue, Kazuhisa Furue, Gaku Tsuji, Takeshi Nakahara

**Affiliations:** 1Department of Dermatology, Graduate School of Medical Sciences, Kyushu University, Maidashi 3-1-1, Higashiku, Fukuoka 812-8582, Japan; ffff5113@gmail.com (K.F.); gakku@dermatol.med.kyushu-u.ac.jp (G.T.); nakahara@dermatol.med.kyushu-u.ac.jp (T.N.); 2Research and Clinical Center for Yusho and Dioxin, Kyushu University, Maidashi 3-1-1, Higashiku, Fukuoka 812-8582, Japan; 3Division of Skin Surface Sensing, Graduate School of Medical Sciences, Kyushu University, Maidashi 3-1-1, Higashiku, Fukuoka 812-8582, Japan

**Keywords:** psoriasis, keratinocytes, interleukin-17A, Th17, ILC3, CCL20, CXCL1, CXCL8, Koebner phenomenon, antimicrobial peptides

## Abstract

The excellent clinical efficacy of anti-interleukin 17A (IL-17A) biologics on psoriasis indicates a crucial pathogenic role of IL-17A in this autoinflammatory skin disease. IL-17A accelerates the proliferation of epidermal keratinocytes. Keratinocytes produce a myriad of antimicrobial peptides and chemokines, such as CXCL1, CXCL2, CXCL8, and CCL20. Antimicrobial peptides enhance skin inflammation. IL-17A is capable of upregulating the production of these chemokines and antimicrobial peptides in keratinocytes. CXCL1, CXCL2, and CXCL8 recruit neutrophils and CCL20 chemoattracts IL-17A-producing CCR6^+^ immune cells, which further contributes to forming an IL-17A-rich milieu. This feed-forward pathogenic process results in characteristic histopathological features, such as epidermal hyperproliferation, intraepidermal neutrophilic microabscess, and dermal CCR6^+^ cell infiltration. In this review, we focus on IL-17A and keratinocyte interaction regarding psoriasis pathogenesis.

## 1. Introduction

Psoriasis is an immune-mediated chronic skin disease characterized by epidermal hyperproliferation, an intraepidermal accumulation of neutrophils, and dermal inflammatory cell infiltrates that are composed of dendritic cells and T cells [1]. It has an estimated global prevalence of 2–4% [1,2]. Males are twice as likely as females to be affected [3,4]. As desquamative erythema can affect any skin site, psoriasis profoundly impairs the patients’ quality of life, treatment satisfaction and adherence, and socioeconomic stability [5,6,7]. The skin lesion usually appears on the sites with frequent trauma such as elbows and knees [8,9]. This injury-induced development of psoriasis is called Koebner phenomenon [8,9]. Environmental factors, such as smoking, also trigger or exacerbate psoriatic lesions [10,11]. Genetic factors are critically involved in the development of psoriasis [12,13]. In addition to skin eruption, approximately 30% of patients with psoriasis manifest psoriatic arthritis [14,15,16,17,18,19]. Psoriasis is also significantly comorbid with other autoimmune diseases, such as bullous pemphigoid [20,21,22,23,24]. Psoriasis is frequently associated with cardiovascular diseases, metabolic diseases, and renal disorders [25,26,27,28,29,30,31,32,33,34,35,36]. Cancer risk is slightly higher in patients with psoriasis [37]. The topical application of steroids and vitamin D3 analogues inhibits psoriatic inflammation and normalizes epidermal differentiation [38,39]. Systemic treatments, such as methotrexate, cyclosporine, phototherapy, and the phosphodiesterase 4 inhibitor apremilast, are useful for patients with extensive lesions [40,41,42,43,44,45]. 

That the tumor necrosis factor-α (TNF-α) and IL-23/IL-17A axes appear to be major drivers in the pathogenesis of psoriasis is underscored by the excellent response of psoriasis to biologics targeting TNF-α, IL-23, and IL-17A, although a difference exists in their efficacy [46,47,48,49,50,51,52,53,54,55,56,57,58,59,60,61]. Anti-TNF-α/IL-23/IL-17A biologics successfully improve psoriatic arthritis [18,56,62,63,64]. Reductions in comorbid cardiovascular events and systemic inflammation have been reported in patients with psoriasis treated with anti-TNF/IL23/IL17 biologics [65,66].

As the clinical response to anti-IL-23/IL-17A biologics seems better than that to anti-TNF-α biologics in psoriasis, the IL-23/IL-17A axis likely plays a more crucial role than the TNF-α axis in the development of psoriasis [67,68,69,70,71]. Under the regulation of IL-23p19, IL-17A-producing CD4^+^ helper T (Th17) cells create a self-amplifying, feed-forward inflammatory response that is markedly augmented in the presence of TNF-α [68,72,73]. In addition, Th17 cells produce high amounts of TNF-α [74,75]. Therefore, IL-17A inhibition by the anti-IL-17A biologic results in early clinical, histopathologic, and molecular resolution of psoriasis [69].

In addition, various murine psoriasis models stress a pivotal role of the IL-23/IL-17A axis in experimental psoriasis [76,77,78,79,80]. Multiple animal studies have indicated that the interaction between IL-17A and keratinocyte is the key issue in the development of psoriasis [76,77,78,79]. In parallel, the importance of IL-17A and keratinocyte interaction is reinforced in patients with psoriasis who are successfully treated with the anti-IL-17A biologic [69]. In this review, we will focus on the multifaceted biological response in keratinocytes stimulated by IL-17A with regard to psoriatic pathogenesis. 

## 2. IL-17A Signaling System

The IL-17 family plays a critical role in the immune response in infectious, inflammatory, autoimmune, and neoplastic disorders [81,82]. The IL-17 family and its receptors are evolutionally ancient, and they are present in species as early as lamprey and sea urchins [83,84]. IL-17 family members comprise IL-17A, IL-17B, IL-17C, IL-17D, IL-17E, IL-17F, and IL-17AF [81,82]. IL-17AF is a hybrid heterodimer of IL-17A and IL-17F. IL-17E is also called IL-25 and is related to the type 2 immune response and allergies [81,82,85]. Among IL-17 family members, IL-17A has been the most strongly implicated in human health and disease. IL-17A is produced from hematopoietic cells, including Th17, CD8+ cytotoxic T cell (Tc17), γδ T cell, natural killer cell, group 3 innate lymphoid cell (ILC3), and “natural” Th17 cell [86,87,88,89], but IL-17B, IL-17C, and IL-25 are preferentially produced from nonhematopoietic cells, including keratinocytes [81,82,90]. Both hematopoietic cells and keratinocytes produce IL-17F [81,82,90]. Keratinocyte-derived IL-17C is capable of stimulating Th17 cells to secrete more IL-17A [91]. 

The IL-17 receptor (IL-17R) is composed of five members: IL-17RA, IL-17RB, IL-17RC, IL-17RD, and IL-17RE [81,82,90]. IL-17A, IL-17F, and IL-17AF all share the same IL-17R comprising IL-17RA and IL-17RC heterodimers [81,82,90]. IL-17C binds to IL-17RA and IL-17RE heterodimers [92]. IL-25 ligates IL-17RA and IL-17RB [81,82,90]. A recent study showed that IL-17A also activates IL-17RA and IL-17RD heterodimers [93] (Figure 1). IL-17B binds IL-17RB, but another heterodimeric IL-17R member remains unidentified [82]. Keratinocytes express both IL-17RA/IL-17RC and IL-17RA/IL-17RD, and the binding of IL-17A induces the transcription of differential gene sets [93]. Initial subcellular events in the ligation of IL-17RA/RC by IL-17A are the recruitment and activation of ACT1 encoded by the *TRAF3IP2* gene, TRAF6 and CARMA2 complexes, and the downstream activation of nuclear factor kappa-light-chain-enhancer of activated B cells (NF-κB) and MAPKs [93,94,95,96,97]. The ligation of IL-17RA/IL-17RC by IL-17A induces the activation of NF-κB, ERK, p38 MAPK, and JNK, while that of IL-17RA/IL-17-RD mainly activates p38 MAPK and JNK and barely affects NF-κB and ERK [93]. In addition, IL-17RA physically and functionally interacts with and transactivates epidermal growth factor (EGFR) [98]. IL-17RD potentially interacts with and transactivates fibroblast growth factor 2 receptor [82,99]. 

In addition to the above-mentioned signaling cascades, IL-17A activates various other signal molecules including signal transducer and activator of transcription 3 (STAT3) in keratinocytes [100]. STAT3 is a very crucial signaling molecule in the development of psoriasis because transgenic mice with keratinocytes expressing a constitutively active Stat3 (K5.Stat3C mice) develop a skin phenotype either spontaneously, or in response to wounding, that closely resembles psoriasis [101]. Moreover, a STAT3 inhibitor STA-21 inhibits the generation of skin lesion in these psoriatic mice [102]. IL-17A is known to activate STAT3 via receptor-interacting protein 4 (RIP4) activation and upregulates the CCL20 expression [103]. IL-17A also upregulates keratin 17 expression via STAT1 and STAT3 activation [104]. IL-6 and IL-22 also play a synergistic role in development of psoriasis with IL-17A [68]. Notably, both IL-6 and IL-22 are potent STAT3 activators [105]. In accordance, biological or natural molecules such as indirubin and its derivatives useful for inactivating STAT3 exhibit therapeutic potential for psoriasis [106] (Figure 2). It reveals that IL-17 and IL-22 promote keratinocyte stemness and potentiate its regeneration [107]. IL-6 is produced from keratinocytes in response to IL-17A [108]. IL-22 is produced from Th17/22 cells, Th22 cells, and other immune cells [109,110].

In humans, impairment of the IL-17 signal causes infectious diseases, especially by *Candida albicans*, which is a ubiquitous fungus and commensal yeast of the intestines and skin [96]. Notably, deficiency of the *IL17RA*, *IL17RC*, *IL17F*, or *TRAF3IP2* genes is implicated in chronic mucocutaneous candidiasis disease (CMCD), which is characterized by recurrent or persistent infection affecting the nails, skin, and oral and genital mucosae caused by the *Candida* species, often *C. albicans* [96,111,112,113]. Impairment of the IL-17 signal is evident in other immunocompromised inborn errors, including autosomal-dominant hyper IgE syndrome, autosomal dominant *STAT1* gain-of-function, autosomal-recessive autoimmune polyendocrinopathy-candidiasis-ectodermal dystrophy (APECED), autosomal-recessive *CARD9* deficiency, *IL12RB1* deficiency, *IL12B* deficiency, and *RORC* deficiency [96]. However, these inborn errors seem to exhibit more complicated immune defects beyond IL-17 dysfunction and manifest CMCD together with other types of infection, including *Mycobacterium*, *Staphylococcus*, and viral disorders [96]. Of note, mice lacking *Traf3ip2* (*Act1*) or *Il17ra* manifest similar clinical phenotype as human CMCD patients lacking *IL17RA* or *IL17RC* [114,115,116,117]. A recent murine study by Sparber et al. also indicates that Malassezia infection selectively triggers the IL-17A-induced immune response [118]. These findings indicate a crucial role of IL-17A in anti-fungal immunity in humans and mice. 

Mice overexpressing IL-17A in keratinocytes (K14-IL-17A^ind/+^ mice) exhibit severe psoriasiform skin inflammation and vascular dysfunction in conjunction with infiltration of the vasculature by inflammatory myeloid cells [108]. The K14-IL-17A^ind/+^ mice acquire the highest local and systemic IL-17A levels and exhibit a particularly severe psoriasis-like skin phenotype [108]. Homozygous CD11c-IL-17A^ind/ind^ mice and heterozygous CD11c-IL-17A ^ind/+^ mice show a delayed onset of moderate to severe psoriasis-like skin disease associated with reduced amounts of cutaneous IL-17A compared with K14-IL-17A^ind/+^ mice. In agreement with elevated skin and a stepwise increase in systemic IL-17A [119], homozygous CD11c-IL-17A ^ind/ind^ mice develop earlier and more severe skin lesions, as well as more pronounced vascular inflammation and dysfunction than heterozygous CD11c-IL-17A ^ind/+^ mice [76,119]. These experimental models indicate that IL-17A induces psoriasis-like lesions in a dose-dependent manner, irrespective of its cellular source [76,119].

## 3. IL-17A and Psoriasis

In the early 2000s, IL-23 was found to induce the production of IL-17A by activated CD4^+^ T cells, which were later named Th17 cells [74,120]. These cells express RORγt (*RORC*) as the master transcription factor [121] and promoted the anti-infectious defense in the mucosa and skin [96], whereas excessive exposure to IL-23 induces their transformation into autoimmune or autoinflammatory immune cells [122,123]. Accordingly, the intradermal injection of IL-23 induces murine psoriasis-like dermatitis with epidermal acanthosis, neutrophil recruitment, and the infiltration of IL-17-producing T cells [78,79,124]. 

Psoriasis is one of the most typical IL-23/IL-17A-driven human diseases [68,72,73]. Th17 cells, Tc17 cells, IL-17-producing γδ T cells, and ILC3 can all be found in psoriatic skin [89,125,126]. IL-23 is composed of IL-23p40 and IL-23p19 subunits, and the expression of both is upregulated in the lesional skin compared to the nonlesional skin in psoriasis [127]. Dermal dendritic cells and monocytes are major sources of IL-23 production [128]. However, immunohistological staining also proves that IL23p40 and IL23p19 are expressed in normal and psoriatic keratinocytes [127,128,129]. Moreover, the epidermal expression of IL-23p40 and IL-23p19 is stronger in psoriasis lesions than in healthy controls [127,128,129]. Murine keratinocytes also produce biologically active IL-23 [130]. IL-23 can promote keratinocyte growth and histological acanthosis via the JAK2/AKT/STAT3/LMO4 signaling pathway [131]. LIM-domain only protein 4 (LMO4) is a LIM-domain protein that regulates keratinocyte proliferation and differentiation during embryogenesis [131]. In addition, TNF-α is an active stimulator of IL-23 production from keratinocytes in mice and possibly in humans [131]. In parallel, the anti-IL-23p40 antibody (Ustekinumab) [132,133] and anti-IL-23p19 antibodies (guselkumab, risankizumab, and tildrakizumab) [133,134,135] exhibit a high efficacy to psoriasis with a superiority of risankizumab over ustekinumab [133], and guselkumab over the anti-TNF-α antibody adalimumab [134].

The expression of *IL17A*, *IL17F*, and *IL17C* is upregulated in psoriatic lesional skin [136,137,138]. The intradermal injection of IL17C induces epidermal thickening and neutrophil infiltration [139]. As significant efficacy of the anti-IL17C antibody has recently been noted in murine psoriasis and atopic dermatitis [140], its human use has been suggested [141]. However, IL-17A is believed to be the most pathogenic IL-17 family member in psoriasis [68,69,72,73,142]. The lesional skin of psoriasis harbors IL-17A-producing immune cells [89,143], and transcriptomic analysis indicates a clear predominance of IL-17A signals [69,142]. Subsets of IL-17A-producing T cells are reactive to autoantigens, such as LL-37 (cathelicidin), a disintegrin and metalloprotease domain containing thrombospondin type 1 motif-like 5 (ADAMTSL5), keratin 17, and neolipid antigens generated by phospholipase A2 group IV D (PLA2G4D) [20,125,126,144,145,146,147]. IL-17A-dominant immune activation is also detected in the nonlesional skin of psoriasis [148].

There are three commercially available anti-IL17A biologics: secukinumab (fully human IgG1κ anti-IL17A antibody), ixekizumab (fully human IgG4 anti-IL17A antibody), and brodalumab (fully human IgG2 anti-IL17RA antibody). The severity of psoriatic skin lesions is generally evaluated by the Psoriasis Area and Severity Index (PASI) [149]. The efficacy of a therapeutic agent is assessed by the rate of responders who achieve 75%, 90%, and 100% reduction of PASI, namely, PASI75, PASI90, and PASI100, respectively. All three anti-IL-17A biologics exbibit remarkable therapeutic efficacy for psoriasis. The PASI90 scores of secukinumab, ixekizumab, and brodalumab are reported to be as high as 60% to 70% at 12 weeks post treatment [150,151,152]. 

Regarding histology, the psoriatic epidermis is acanthotic and the proliferating capacity of keratinocyte is increased as determined by Ki67 positivity and cytokeratin 16 expression [68,153,154]. In accordance with the accumulation of neutrophils in psoriasis, the gene expression of neutrophilic cytochemokines CXCL1, CXCL2, CXCL8, and IL-36 is upregulated in the lesional skin of psoriasis [68,155,156,157]. Psoriatic keratinocytes produce a large amount of CCL20 [158,159], which is a potent chemoattractant for CCR6^+^ IL-17A-producing Th17 [89,143,160], ILC3 [161,162], Tc17 [163,164], and γδ T cells [78,165]. Interestingly, IL-17A actively upregulates the production of these key cytochemokines as well as proliferative capacity in keratinocytes. 

## 4. IL-17A and Keratinocyte Proliferation/Differentiation

Psoriatic keratinocytes simultaneously exhibit increased proliferation (Ki67^+^ or cytokeratin 16^+^ cell number) and differentiation [involucrin (IVL)^+^ cell number] [69,166,167,168]. Accordingly, IL17A is an active promoter of keratinocyte proliferation and differentiation [168,169,170,171]. In psoriasis patients, the neutralization of IL-17A by ixekizumab [168] or secukinumab [69] significantly decreases histological acanthosis, the number of Ki-67^+^-proliferating keratinocytes, and epidermal cytokeratin 16 expression. In the murine imiquimod-induced psoriasis model, topical imiquimod induces epidermal thickening that is significantly alleviated in *Traf3ip2* (*Act1*)-deficient mice in which IL-17 signaling is blocked [170]. The molecular mechanisms of the IL-17A-mediated acceleration of keratinocyte proliferation and differentiation are not yet fully understood. 

IL-17A alone is likely to be insufficient to evoke a significant inflammatory response and may cooperatively or synergistically accelerate the proinflammatory cascade in combination with other cytokines, such as TNF-α, IL-23, IL-1β, IL-6, IL-22 and transforming growth factor-β (TGF-β) [68,172]. Transcriptomic analysis reveals a clear additive or synergistic gene regulation by IL-17A and TNF-α in human keratinocytes [173]. This gene regulation is likely attributable to two sets of transcription factors: NF-κB and the C/CAAT-enhancer-binding proteins (C/EBP), C/EBPβ, or C/EBPδ [169,173,174]. TNF-α is a strong inducer of active NF-κB, while IL-17A activates C/EBPβ or C/EBPδ and to a lesser extent NF-κB [169,173,174]. Therefore, the IL-17A blockade reduces the expression of these responsive genes to a greater extent than TNF-α inhibition [168].

The C/EBP family members are highly expressed transcription factors in epidermal keratinocytes and sebocytes in mice and humans and accelerate their differentiation [175]. The expression of C/EBPβ protein is located in the upper spinous layer and is strongly upregulated in the lesional skin of psoriasis [169]. Together with the elevated gene expression of *CEBPB*, the expression of keratinocyte-terminal differentiation genes, such as *IVL*, *FLG2*, and *TGM1*, is upregulated in the lesional skin of psoriasis [169]. In the promoter region of the *IVL* gene, there is a binding site for C/EBP, and the C/EBP transcription factor is necessary for the appropriate and continuous production of IVL protein [176]. These results suggest that IL-17A accelerates keratinocyte differentiation by increasing C/EBPβ protein in keratinocytes [168,169,173]. 

Regarding keratinocyte proliferation, IL-17A stimulates keratinocytes to produce IL-19 [173,177]. The combination of IL-17 and TNF-α results in the synergistic expression of IL-19 in keratinocytes. IL-17 alone promotes IL-19 expression by approximately 1.79-fold, and TNF-α alone slightly reduces IL-19 expression, whereas a combination of IL-17 and TNF-α promotes expression by 54.6-fold [173,177]. IL-19 itself promotes keratinocyte migration but not proliferation [178]. However, it activates fibroblasts to produce keratinocyte growth factor, which upregulates keratinocyte proliferation [178]. Additionally, the upregulation of cell cycle-related genes, such as CCNE1, CDCA5, and CDCA25A, suggests a direct contribution of IL-17 to epidermal KC proliferation [169]. The expression of cytokeratin 16 is upregulated in the lesional epidermis of psoriasis patients and in keratinocytes incubated with IL-17A [69,168,179]. The expression of cytokeratin 16 increases the proliferative capacity of keratinocytes via EGFR phosphorylation [180].

Other studies also underpin a possibility that IL-17A activates EGFR [98,181,182]. Chen et al. have demonstrated that IL-17A transactivates EGFR in keratinocyte stem cells [98]. IL-17A accelerates the enzymatic cleavage of amphiregulin, which activates EGFR in airway epithelial cells [181]. In the lesional skin of psoriasis, the expression of EGFR ligands, such as heparin-binding EGF, transforming growth factor-α, and amphiregulin, is overexpressed [183]. Consistent with these notions, the inhibition of EGFR by erlotinib or cetuximab successfully improves severe psoriasis [184,185,186,187]. These studies suggest that EGFR activation may partly explain the IL-17A-mediated upregulation of keratinocyte proliferation.

## 5. IL-17A and Cyto/Chemokines in Keratinocytes

In keratinocytes, IL-17A upregulates the expression of various psoriasis-related cyto/chemokines and antimicrobial peptides, such as CXCL1, CXCL8, IL-36G, CCL20, IL-19, IL-17C, S100A7, S100A8, S100A9, LL-37, and defensin β 4A (DEFB4A) [157,169,173,188,189,190]. The expression of these molecules is upregulated in the lesional skin of psoriasis and is downregulated to normal levels by biologics targeting TNF-α [128] or IL-17A [69,168]. Some are upregulated even in the nonlesional skin of patients with psoriasis [148]. The IL-17A-induced upregulation of these molecules is further amplified in the presence of TNF-α and IL-19 [173,177].

CXCL1 and CXCL8 are potent chemoattractants for neutrophils [191,192,193]. IL-36G induces CXCL1 and CXCL8 expression in an autocrine fashion in keratinocytes and recruits neutrophils [194]. Full-length IL-36G is cleaved by cathepsin G released from infiltrated neutrophils, and the cleaved IL-36G exhibits a more potent functional activity than the full-length one [192,195,196]. Moreover, IL-17A upregulates IL-36G production more potently in human psoriasis-derived keratinocytes than in healthy keratinocytes [197]. Therefore, the IL-17A, IL-36G, CXCL1/CXCL8, and neutrophils form a feed-forward vicious cycle, and an intraepidermal neutrophilic microabscess (Munro’s microabscess or Kogoj’s spongiform pustule) may develop [156,198,199]. The pathogenic significance of IL-36 is stressed more in pustular rather than plaque psoriasis [156,198,199]. In parallel, inhibition of the IL-36 pathway is efficacious for the treatment of pustular psoriasis [200]. 

While most chemokines redundantly bind to multiple receptors, CCL20 has only one known receptor, CCR6 [191]. CCR6 is expressed on dendritic cells [159] and IL-17-producing immune cells, including Th17 [89,143,201], ILC3 [161,162], Tc17 [163,164], and γδ T cells [78,165]. CCR6 is now considered a representative marker for Th17 cells [202,203]. The human psoriatic epidermis expresses abundant CCL20 with the dermal infiltration of CCR6^+^ dendritic cells and skin-homing T cells [143,158,159]. Although dermal dendritic cells and T cells express CCL20 in the psoriatic dermis [159], the expression of CCL20 is largely confined to the psoriatic epidermis, suggesting that epidermal keratinocytes are the major source of CCL20 production in psoriatic lesions [158,159]. CCL20 is constitutively expressed in cultured keratinocytes [201,204], and its production is upregulated by TNF-α and IL-17A [182,204]. Scratch injury upregulates the gene expression of *CCL20*, *CXCL8*, and *IL36G,* which may be related to the scratch-induced Koebner phenomenon frequently observed in patients with psoriasis [201]. Preclinical studies have revealed that the humanized anti-CCR6 antibody efficiently inhibits the cutaneous infiltration of CCR6^+^ T cells in human and murine models [205,206]. The development of small molecule inhibitors against CCR6 is also ongoing [207,208]. Targeting the CCL20/CCR6 axis may be a potential therapeutic strategy for the treatment of psoriasis. 

## 6. IL-17A and Anti-Microbial Peptides in Keratinocytes

S100A7, S100A8, S100A9, LL-37, and DEFB4A are anti-microbial peptides, and their expression is upregulated by IL-17A in keratinocytes [189,209,210]. S100A7 (psoriasin) has a multifaceted role in keratinocyte pathophysiology, including wound healing, keratinocyte differentiation, nucleocytoplasmic transport, and chemotaxis for CD4^+^ T cells, neutrophils, and monocytes [211]. The production of S100A7 is augmented in the presence of IL-19 [212], IL-36G [188], and TNF-α [213]. S100A8, S100A9, and S100A12 are alternatively known as calgranulin A, B, and C, respectively, and exhibit antimicrobial activity [214]. All are elevated in psoriatic plaque and decreased by treatment with anti-IL-17A antibody secukinumab [69,168,215]. The S100 family members primarily form homodimers or higher-order oligomers, but S100A8 and S100A9 uniquely form heterodimeric complexes, which are known as calprotectin [214]. The chelation of Zn2^+^ and Mn2^+^ by extracellular S100A8/A9 is a proposed mechanism of antimicrobial activity [214]. S100A12 also has a chemotactic activity for mast cells and monocytes [216]. Pure human S100A8/A9 has broad spectrum antimicrobial activities against microorganisms, including *Capnocytophaga sputigena*, *C. albicans*, *Escherichia coli, Staphylococcus aureus, S. epidermis*, *Borrelia burgdorferi*, and *Listeria monocytogenes*, in vitro, and S100A12 has antimicrobial activity against filarial parasites [214]. The receptor for advanced glycation end-products (RAGE) may serve as a common receptor for S100 proteins, including S100A7, S100A8, S100A9, and S100A12 [214]. S100A8/S100A9 augments the production of CXCL1, CXCL2, CXCL3, CXCL8, CCL20, IL-6, and TNF-α in keratinocytes [217], and it enhances keratinocyte proliferation [217]. The transcriptional co-activator IκBζ, encoded by the *NFKBIZ* gene, plays a critical role in IL-17A-, IL-17F-, and IL-17A/F-mediated signaling, such as the gene expression of *S100A7* and *CCL20* [218,219,220]. 

LL-37 is an antimicrobial peptide of human cathelicidin that is produced when keratinocytes are injured by a broad range of bacteria, viruses, and fungi [221,222]. In addition to antimicrobial peptide activity, LL-37 exhibits “alarmin” function, affects adenosine triphosphate-receptor P2X7 and Toll-like receptor (TLR) signaling, and EGFR transactivation or intracellular Ca2+ mobilization [223,224,225]. The released LL-37 binds to the infiltrated neutrophils [221]. Neutrophils are a rich source of extracellular DNA due to their neutrophil extracellular traps [226]. Upon stimulation with complexes of host DNA and LL-37, plasmacytoid dendritic cells produce large amounts of IFN-α [227]. Notably, LL-37 induced the proliferation of circulating CD3^+^ T cells in 24 out of 52 patients with psoriasis (46%) [144]; therefore, LL-37 is effective for autoantigens. In total, 50 LL-37-reactive CD3^+^ T cells, including both CD4^+^ and CD8^+^ T cells, express the skin-homing receptor cutaneous lymphocyte antigen [144]. LL-37 peptides bind to HLA-DR in dendritic cells and are presented to CD4^+^ T cells, while LL-37 peptides and the HLA-C*0602 complex activate CD8^+^ T cells [144,228]. The majority of LL-37-reactive CD3^+^ T cells produce IL-17, and the capacity of their IL-17 production is associated with disease severity [144]. Interestingly, the LL-37-specific IL-17-producing T cells are exclusively CD4^+^, whereas the LL-37-specific CD8^+^ T cells do not produce IL-17 [144]. A recent study by Takahashi et al. revealed that LL-37 can bind to self-RNA and stimulate macrophages to produce IL-6 via a scavenger receptor [229].

DEFB4A is highly expressed in psoriasis plaques and is the most psoriasis-specific antimicrobial peptide [230,231]. In contrast, DEFB4A is expressed at negligible or low levels in normal skin and skin lesions of eczema [230,231,232]. The expression of DEFB4A is upregulated by IL-17A and synergistically by IL-17A and TNF-α [215,220]. The serum levels of DEFB4A are highly specific biomarkers for disease activity in patients with psoriasis [215]. The number of neutrophil extracellular traps increase in psoriasis and upregulate the expression of DEFB4A [226]. Notably, DEFB4A is a functional (non-chemokine) ligand for CCR6 and feasibly attracts Th17 cells [233,234]. 

Although the biological implications of increased antimicrobial peptides in psoriasis remain obscure, they are intimately associated with IL-17A-rich milieu. Therefore, the upregulated expression of these antimicrobial peptides is rapidly normalized by the neutralization of IL-17A by ixekizumab [168] or secukinumab [69]. 

## 7. Conclusions

IL-17A is a multifunctional cytokine produced from adaptive and innate immune cells, such as Th17 and ILC3s. It orchestrates and promotes the peripheral tissue defense system against microbial insult, especially fungal infection. Psoriasis is a major inflammatory skin disease in which the interaction between IL-17A and epidermal keratinocytes plays a critical pathogenic role. IL-17A stimulates the proliferation of keratinocytes. Keratinocytes also produce a variety of antimicrobial peptides and cytochemokines in response to IL-17A. The antimicrobial peptides further exacerbate skin inflammation. CCL20 produced from IL-17A-stimulated keratinocytes recruits IL-17A-producing Th17 cells and ILC3s and accelerates the feed-forward vicious cycle, which causes fully developed psoriasis. This pathogenetic scheme has been verified with a high clinical efficacy of anti-IL-17A biologics. Therefore, psoriasis is considered an excellent human model of how IL-17A works with target peripheral tissues, and it provides in-depth insight into human autoinflammatory diseases.

## Figures and Tables

**Figure 1 ijms-21-01275-f001:**
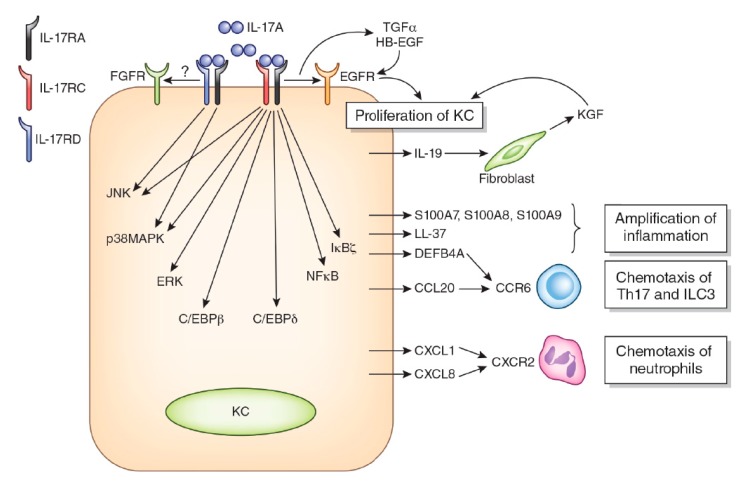
Simplified effects of anti-interleukin 17A (IL-17A) on keratinocyte (KC) with regard to psoriasis pathogenesis. IL-17A homodimers bind to IL-17 receptor A (IL-17RA) and IL-17RC or IL-17RA and IL-17RD heterodimers. The ligation of IL-17RA/IL-17RC activates epidermal growth factor receptor (EGFR) directly or by transforming growth factor-α (TGF-α) and heparin-binding EGF-like growth factor (HB-EGF) and promotes keratinocyte proliferation. The ligation of IL-17RA/IL-17RC activates various signal transduction molecules, including ERK, p38 MAPK, JNK, nuclear factor kappa-light-chain-enhancer of activated B cells (NFκB), IκBζ, C/CAAT-enhancer-binding protein β (C/EBPβ), and C/EBPδ. In contrast, the ligation of IL-17RA/IL-17RD preferentially activates JNK and p38 MAPK pathways. IL-17RA/IL-17RD is estimated to transactivate fibroblast growth factor receptor (FGFR); however, this is not conclusive. IL-17RA/IL-17RC signaling stimulates KCs to produce IL-19, which induces the production of keratinocyte growth factor (KGF) from fibroblasts. KGF also enhances the proliferation of KCs. IL-17A also induces the production of antimicrobial peptides, including S100A7, S100A8, S100A9, LL-37, and defensin β 4A (DEFB4A). These antimicrobial peptides amplify the local inflammatory process. Chemokines, such as CCL20, CXCL1, and CXCL8, are also produced from keratinocytes by IL-17RA/IL-17RC ligation. CCL20 is a key chemokine for the recruitment of CCR6^+^ Th17 cells and group 3 innate lymphoid cells (ILC3). These CCR6^+^ cells produce large amounts of IL-17A. DEFB4A also exhibits a chemotactic activity by binding to CCR6. CXCL1 and CXCL2 are potent chemoattractants for CXCR2^+^ neutrophils. Therefore, IL-17A is associated with all of the histopathologic features of psoriasis.

**Figure 2 ijms-21-01275-f002:**
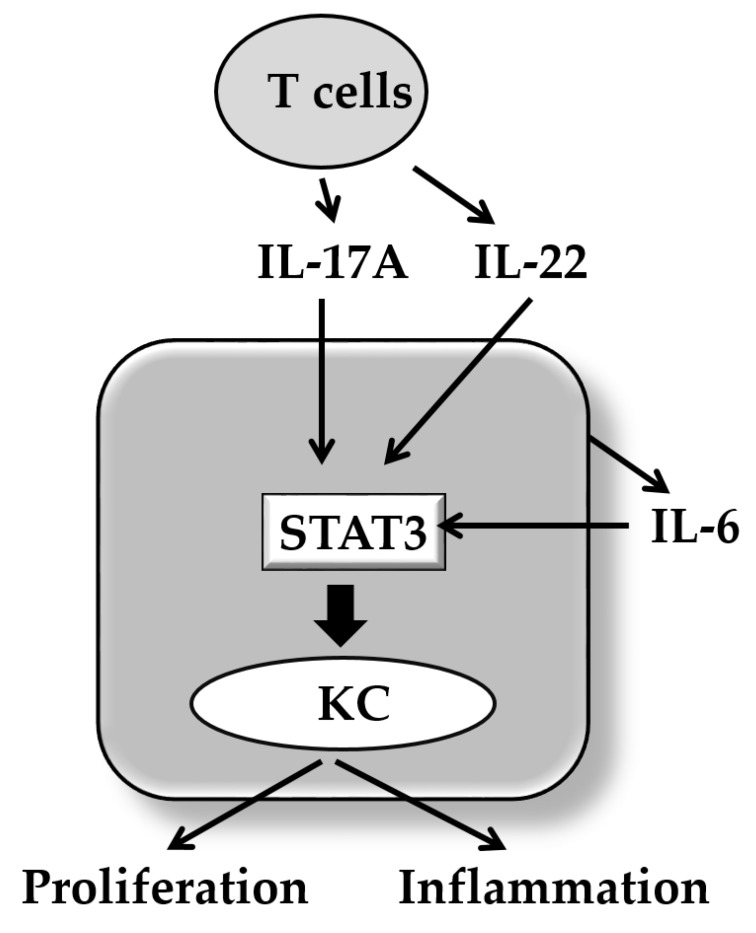
Pivotal role of signal transducer and activator of transcription 3 (STAT3) in psoriasis. The activation of STAT3 promotes keratinocyte (KC) proliferation and inflammatory response. IL-17A and IL-22 induce the STAT3 activation. IL-6 produced from KC also induces STAT3 activation.

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
