# Peer review of "Interleukin-17A and Keratinocytes in Psoriasis"

_ijms, 2020, doi:10.3390/ijms21041275_

Round 1

Reviewer 1 Report

The topic is very interesting. The authors have made a very thorough analysis of the literature. However, they should limit references to the most relevant.

General information in the introduction is random and unrelated to each other. In my opinion these sentences are not directly related to the topic and have no significant meaning in the introduction.

For example: The involvement of the chemical sensor aryl hydrocarbon receptor is also suggested in the pathophysiology of psoriasis [12]In addition to skin eruption, approximately 30% of patients with psoriasis manifest psoriatic arthritis [14-19]. Psoriasis is also significantly comorbid with other autoimmune diseases, such as bullous pemphigoid [20-24]….

The names of medicines, e.g. ixekizumab should be in lowercase. Maybe it would be worth enriching this manuscript with a table summarizing the effect of IL-17A on the biology of keratinocytes in psoriasis.

Author Response

Reply to the Reviewer 1

General information in the introduction is random and unrelated to each other. The literature should be corrected and limited to the main items.

→ Thank you very much for your critical comments. We agree with your comment. According to your comment, we deleted some sentences and amended the introduction as follows adding new reference Ref #12.

Section 1. Introduction

 “This injury-induced development of psoriasis is called Koebner phenomenon [8,9]. Environmental factors, such as smoking, also trigger or exacerbate psoriatic lesions [10,11]. Genetic factors are critically involved in the development of psoriasis [12, 13].”

Thank you so much again for your helpful comment. We hope the revised version will be suitable for publication in IJMS.

Reviewer 2 Report

In this paper the authors present a review of data regarding the role of Interleukin-17A in autoinflammatory skin disease such as psoriasis. They dissect the pathway by which this interleukin activate the inflammatory response. specifically they describe its role in the activation of signal transduction molecules, like ERK, p38 MAPK, JNK, NFκB, IκBζ.

However the signalling activated by IL-17A seems to be more complex than the presented in the cartoon of figure 1. As it is been known from the literature IL-17 is not the only interleukin involved in this process. A direct link between IL-17, IL-6 and STAT3 activation is widely described, and should be analysed in this review. Moreover also the involvemt of IL-22 in the activation of STAT3 pathway has been described, and alongside biological drugs, also natural molecules has been presented as inactivator of STAT3 pathway. In this context the possibility tu use biological or natural molecules useful to inactivate also this pathway should be treated. especially focusing the attention on the fact that one possibility is to use molecules able to drive keratinocyte differentiation, that is a way to impair the hyperproliferation due to the inflammation.

-The review contain a single picture that fail to completely elucidate the processe described or that should be described.

Author Response

Reply to the Reviewer 2

In this paper the authors present a review of data regarding the role of Interleukin-17A in autoinflammatory skin disease such as psoriasis. They dissect the pathway by which this interleukin activate the inflammatory response. specifically they describe its role in the activation of signal transduction molecules, like ERK, p38 MAPK, JNK, NFκB, IκBζ. However the signalling activated by IL-17A seems to be more complex than the presented in the cartoon of figure 1. As it is been known from the literature IL-17 is not the only interleukin involved in this process. A direct link between IL-17, IL-6 and STAT3 activation is widely described, and should be analysed in this review. Moreover also the involvemt of IL-22 in the activation of STAT3 pathway has been described, and alongside biological drugs, also natural molecules has been presented as inactivator of STAT3 pathway. In this context the possibility to use biological or natural molecules useful to inactivate also this pathway should be treated. especially focusing the attention on the fact that one possibility is to use molecules able to drive keratinocyte differentiation, that is a way to impair the hyperproliferation due to the inflammation.

 → Thank you very much for your valuable comment. We agree with your suggestion. According to your comment, we added the following sentences by adding appropriate references.

Section 2. IL-17A signaling system

“In addition to the above mentioned signaling cascades, IL-17A activates various other signal molecules including STAT3 in keratinocytes [100]. STAT3 is a very crucial signaling molecule in the development of psoriasis because transgenic mice with keratinocytes expressing a constitutively active Stat3 (K5.Stat3C mice) develop a skin phenotype either spontaneously, or in response to wounding, that closely resembles psoriasis [101]. Moreover, a STAT3 inhibitor STA-21 inhibits the generation of skin lesion in this psoriatic mice [102]. IL-17A is known to activate STAT3 via receptor-interacting protein 4 (RIP4) activation and upregulates the CCL20 expression [103]. IL-17A also upregulates keratin 17 expression via STAT1 and STAT3 activation [104]. IL-6 and IL-22 also play a synergistic role in development of psoriasis with IL-17A [68]. Notably both IL-6 and IL-22 are potent STAT3 activator [105]. In accordance, biological or natural molecules such as indirubin and its derivatives useful for inactivating STAT3 exhibit therapeutic potential for psoriasis [106] (Figure 2). It reveals that IL-17 and IL-22 promote keratinocyte stemness and potentiate its regeneration [107]. IL-6 is produced from keratinocytes in response to IL-17A [108]. IL-22 is produced from Th17/22 cells, Th22 cells and other immune cells [109,110]. ”

-The review contain a single picture that fail to completely elucidate the processe described or that should be described.

→ Thank you for your helpful comment. According to your comment, we added Figure 2 in the revised article and stressed the importance of STAT3 signal.

Thank you so much again for your helpful comment. We hope the revised version will be suitable for publication in IJMS.

Round 2

Reviewer 2 Report

The authors presented a revised and improved version of the paper, according the previous referee requests.

They included a new paragraph regarding STAT3 involvement in the pathogenesis of psoriasis.

Although they included fundamental references (Sano et al 2005), only indirubin has been presented as example of natural compound. This compound is produced mainly as a bioproduct of bacterial metabolism, and present in indigo naturalis, but the landscape of natural compound offer a number of other interesting molecules. Moreover the picture 2 is very simplistic, since it do not provide correct information about STAT3 pathway, such as the modifications that are important for its activity. These modifications include modification by Sirtuin, phosphorylation, dimerization and nuclear translocation.